# Could forage peanut in low proportion replace N fertilizer in livestock systems?

**Vanessa Z. Longhini**[1]*, **Abmael S. Cardoso**[1], **Andressa S. Berça**[1], **Robert M. Boddey**[2], **Ricardo A. Reis**[1], **José C. B. Dubeux, Jr.**[3], **Ana C. Ruggieri**[1]

**1** Department of Animal Science, São Paulo State University, Jaboticabal, SP, Brazil, **2** Embrapa Agrobiologia, Antiga Rodovia Rio-São Paulo, Seropédica, RJ, Brazil, **3** University of Florida, North Florida Research and Education Center, Marianna, FL, United States of America

* vanessa.longhini@unesp.br

**Data Availability Statement:** All relevant data are within the paper and its Supporting Information file.

## Abstract

Palisadegrass [*Urochloa brizantha* (Hochst. ex A. Rich.) R. D. Webster cv. Marandu] is widely used in Brazil and is typically managed with little or no N fertilizer, which often leads to pasture decline in the long-term. The current relationship between beef price and fertilizer cost in Brazil does not favor fertilizer use in pastures. Legume inclusion is an alternative to adding fertilizer N, but often legumes do not reach a significant proportion (> 30%) in pasture botanical composition. This study evaluated herbage responses to N inputs and pasture species composition, under intermittent stocking. Treatments included palisadegrass-forage peanut (*Arachis pintoi* Krapov. & W.C. Greg. cv. Amarillo) mixture (mixed), unfertilized palisadegrass (control), and palisadegrass fertilized with 150 kg N ha$^{-1}$ yr$^{-1}$ (fertilized). Treatments were applied over two rainy seasons with five growth cycle (GC) evaluations each season. Response variables included herbage biomass, herbage accumulation, morphological components, total aboveground N of forage peanut (TAGN$_{FP}$), and contribution of biological N$_2$ fixation (BNF). Herbage biomass was greater for fertilized palisadegrass [5850 kg dry matter (DM) ha$^{-1}$] than for the palisadegrass-forage peanut mixture (3940 kg DM ha$^{-1}$), while the unfertilized palisadegrass (4400 kg DM ha$^{-1}$) did not differ from the mixed pasture. Nitrogen fertilizer increased leaf mass of palisadegrass (2490 kg DM ha$^{-1}$) compared with the control and mixed treatments (1700 and 1310 kg DM ha$^{-1}$, respectively). The contribution of BNF to the forage peanut ranged from 79 to 85% and 0.5 to 5.5 kg N ha$^{-1}$ cycle$^{-1}$. Overall, benefits from forage peanut were minimal because legume percentage was less than 10%, while N input in the system by N-fertilizer increased palisadegrass herbage biomass.

## Introduction

In Brazil, grasses of the genus *Urochloa* cover an estimated 80 million ha. Marandu palisadegrass [*Urochloa brizantha* (Hochst. ex A. Rich.) R. D. Webster cv. syn. *Brachiaria brizantha* (Hochst. ex A. Rich.) Stapf.], the most used palisadegrass cultivar, is estimated to cover 56% of this area [1]. Low-input systems prevail in Brazilian livestock operations with minimal

**Funding:** The authors were supported by São Paulo Research Foundation – FAPESP (VZL: grant # 2016/11086-1; ASC: grant# 2017/11274-5, and RAR: thematic grant # 2015/16631-5, https://fapesp.br/) and by Cientista de Nosso Estado for author RMB from the Rio de Janeiro State Research Foundation – FAPERJ (http://www.faperj.br/). This work was funded by National Council for Scientific and Technological Development – CNPq (RMB: grant # 404169/2013-9, https://www.gov.br/cnpq/pt-br). The funders had no role in study design, data collection and analysis, decision to publish, or preparation of the manuscript.

**Competing interests:** The authors have declared that no competing interests exist.

applications of limestone, gypsum, or N fertilizers. The combination of low soil fertility, lack of N fertilizer application, and overgrazing results in pasture degradation or pasture decline [2].

Nitrogen is a key element to improve pasture productivity [3] and many warm-season $C_4$ grasses respond positively to N fertilization by increased herbage accumulation and improved crude protein [4]. However, this response can vary with the environment, species, and management practices. Delevatti [4] reported that herbage accumulation of palisadegrass increased with increasing N fertilizer applications from 90 to 270 kg ha$^{-1}$, but there was a diminished increase in herbage accumulation per unit of N applied. In this case, tripling the amount of N applied increased herbage accumulation by only 18% and increased potential for negative environmental impacts. Nitrogen fertilizer is manufactured using fossil fuel, which results in a significant carbon footprint [5]. Furthermore, N fertilizer may be lost as in gaseous forms such as nitric oxide (NO), nitrogen dioxide ($NO_2$), nitrous oxide ($N_2O$), di-nitrogen ($N_2$), ammonia ($NH_3$), or via leaching of nitrate ($NO_3^-$) [6, 7]. Several of these processes contribute to climate change (e.g., $N_2O$ emission) or increase $NO_3^-$ concentrations in surface water bodies and groundwater.

Integrating legumes into grazing systems is an alternative to mineral N fertilizer applications that may reduce negative environmental impacts [8]. Legumes add N to the system via symbiotic biological $N_2$ fixation (BNF) between legume and rhizobium bacteria in root nodules [9]. Legumes can also reduce production costs compared with N fertilizer [8, 10].

The use of grass-legume mixed pastures in tropical regions is less commonplace than in temperate regions. In the Southeastern United States, mixing grasses and legumes is a more common practice [11] that requires further study [12, 13]. In Brazil, few studies have evaluated grass-legume mixtures under grazing. Challenges to legume use include poor establishment and persistence compared with grasses [10, 14, 15]. Forage peanut (*Arachis pintoi* Krapov. & W.C. Greg. cv. Amarillo) is a legume that has good potential for mixtures in Brazil because it persists well when mixed with fast-growing grasses such as *Urochloa* [16, 17]. A stoloniferous and prostrate growth habit makes the forage peanut ideal for grazing because its meristems are protected from grazing animals [18]. Under tropical conditions, Tamele [15] reported that palisadegrass managed at a 20-cm canopy height improved forage peanut establishment in mixed systems. However, further research is necessary to determine if N from forage peanut grown in mixture with a grass is able to substitute for synthetic N fertilizer. The proportion of the legume in the mixture is important to determine the benefits of the association. According to Thomas [19], proportions between 20–45% in the herbage biomass are sufficient to supply N to the agroecosystem. However, little has been done in terms of determining at which level the legume becomes important, or even if, at low participation, the legume is still able to benefit the overall system.

We hypothesized that the forage peanut mixed with non-N fertilized grass would benefit the herbage responses through BNF, compared with unfertilized grass in a monoculture. However, the proportion of forage peanut in the sward in the present study was low. Thus, we tested whether there is a meaningful and measurable contribution of the N from forage peanut even at minimal participation and whether this contribution could be close to those systems using synthetic N fertilizer. Therefore, this study evaluated the performance of palisadegrass under three nitrogen management strategies (unfertilized, N fertilized, or mixed with the legume forage peanut), aiming to identify how different N inputs may affect grass responses.

## Materials and methods

### Ethical approval

The procedures involving animals were approved by the Institutional Animal Care and Use Committee at São Paulo State University, SP, Brazil (No. 10356/14-CEUA).

## Experimental site

This study was conducted at Sao Paulo State University, Jaboticabal, Sao Paulo, Brazil (21°14'S and 48°17'W; 586 m asl) during two rainy seasons (November 2016 to March 2017 and December 2017 to April 2018). The soil at the experimental site is classified as an Oxisol [20]. Soil samples (0- to 20-cm depth) collected in 2014, 2017, and 2018 indicated the following means: pH (CaCl$_2$) 5.3; organic matter 32.4 g kg$^{-1}$; cation exchange capacity 74.8 mmolc dm$^{-3}$; P (ion-exchange resin extraction method) 10.9 mg dm$^{-3}$; Mehlich-1 extractable Ca 28.3 mmolc dm$^{-3}$; Mehlich-1 extractable Mg 9.7 mmolc dm$^{-3}$; Mehlich-1 extractable K 4.2 mmolc dm$^{-3}$; base saturation 561 g kg$^{-1}$ [21]. Soil texture was identified as clay, with 340 g kg$^{-1}$ sand, 140 g kg$^{-1}$ silt and 520 g kg$^{-1}$ clay [22]. The total annual rainfall in this area is 1314 mm and mean annual air temperature is 22.7°C (2010–2018). Rainfall and temperature data from the experimental site were extracted from a dataset belonging to the Agrometeorology section of the Department of Exact Sciences (Fig 1).

## Treatments and experimental design

There were three treatments (nitrogen management strategies, NM) with four replications, arranged in a completely randomized design with plot size of 20 × 60 m (Fig 2). Treatments were 1) Control: Marandu palisadegrass in a monoculture, unfertilized (without N fertilizer or legume); 2) Fertilized: Marandu palisadegrass fertilized with 150 kg N ha$^{-1}$ year$^{-1}$ (urea), split into three equal applications; and 3) Mixed: Marandu palisadegrass mixed with forage peanut cv. Amarillo and no N fertilizer. These treatments were evaluated for two rainy seasons, with five growth cycles (GC) within each season (Table 1). Nitrogen was applied to the fertilized

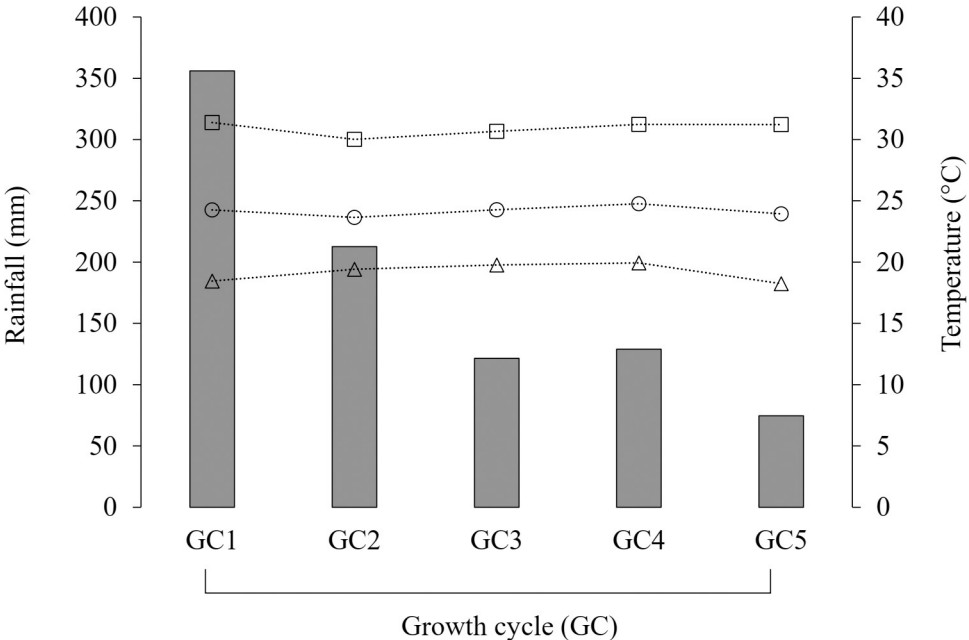

**Fig 1. Cumulative rainfall and mean air temperatures (maximum, mean, and minimum) for five growth cycles (GC) in Jaboticabal, Brazil (Nov/2016-Mar/2017 and Dec/2017-Apr/2018).**

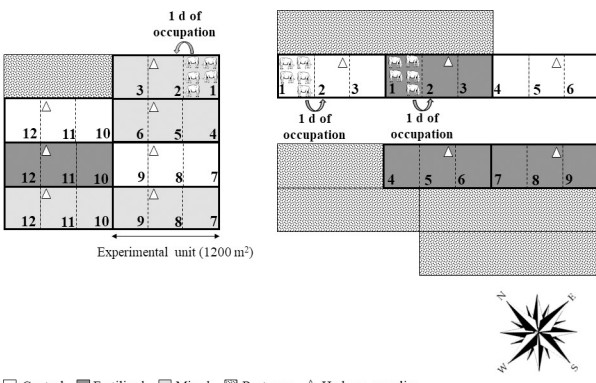

**Fig 2. Schematic representation of the experimental design.** Three treatments with four replications, arranged in a completely randomized design. Experimental unit of $20 \times 60$ m ($1200$ m$^2$). Control: Marandu palisadegrass in a monoculture (without N fertilizer or legume); Fertilized: Marandu palisadegrass fertilized with 150 kg N ha$^{-1}$ year$^{-1}$ (urea split into three equal applications); and Mixed: Marandu palisadegrass mixed with forage peanut cv. Amarillo and no N fertilizer. Intermittent stocking method with grazing periods lasting 3 d, or 1 d for each paddock. Palisadegrass pasture was used as a reserve area to hold the animals during the rest periods.

treatment in November, January, and March of each rainy season (50 kg N ha$^{-1}$ per application).

## Plot establishment

In March 2014, established 'Tifton 85' bermudagrass (*Cynodon* spp.) was sprayed with 2880 g a.i. ha$^{-1}$ of glyphosate [isopropylamine salt of N- (phosphonomethyl) glycine] in order to eradicate the existing vegetation. In October 2014, glyphosate plus 402 g a.i. ha$^{-1}$ 2,4-D amine (2,4-dichlorophenoxyacetic acid) were applied, and subsequently, a crusher and chopper (Tritton, model 1800) were used to remove dead residue. Grass monoculture was seeded subsequently with a row spacing of 0.45 m and a seeding rate of 10 kg ha$^{-1}$ of pure live seed using a no-till seed drill (Semeato, model SHM 11–13). In the mixed treatment, forage peanut was simultaneously sown between rows of palisadegrass at a seeding rate of 8 kg ha$^{-1}$ of pure live seed using a seed drill (Jumil, model 2640 PD Exacta). After seeding, all paddocks received 60 kg P, 20 kg S, 50 g Mo, 1 kg Cu and 1 kg B ha$^{-1}$, following recommendations of Werner [23]. At the beginning of the grazing trial in both years, 40 kg P ha$^{-1}$ (equivalent to 222.2 kg P$_2$O$_5$ ha$^{-1}$) and 50 kg K ha$^{-1}$ (equivalent to 83.3 kg K$_2$O ha$^{-1}$) were applied. The grass-legume mixed pastures were not fertilized with N.

Before seeding in 2014, the three treatments were randomized and allocated into 12 experimental units (1200 m$^2$ each). Each experimental unit was split into three paddocks of 400 m$^2$.

## Grazing management

In September of each year, crossbred dairy heifers were used to stage the plots. The grazing method was intermittent stocking with grazing periods lasting 3 d with 1 d for each paddock (Fig 2). The mob stocking technique was used for pasture defoliation [24] with a minimum of five crossbred dairy heifers ($311 \pm 6$ kg and $234 \pm 4$ kg, 2016/2017 and 2017/2018, respectively) for grazing. Additional heifers were 'put-and-take' as needed to achieve the target stubble height of 15 cm within one day of occupation of a given paddock. The same animal group was used for the three paddocks within an experimental unit.

Grazing for all treatments initiated when one of the treatments reached 95% light interception (LI) [25] as determined using a plant canopy analyzer LAI-2200 (LI-COR®, USA). The

**Table 1. Sequence of events over two production seasons.**

|  | Initial | Final | Rest period [a] |
|---|---|---|---|
| Growth cycle (GC) | Date | | days |
| Growing season (2016/2017) | | | |
| Staging grazing | 07 Sept 2016 | 18 Sept 2016 | - |
| GC1 | 19 Nov 2016 | 30 Nov 2016 | 62 |
| GC2 | 10 Dec 2016 | 21 Dec 2016 | 21 |
| GC3 | 09 Jan 2017 | 20 Jan 2017 | 30 |
| GC4 | 06 Feb 2017 | 17 Feb 2017 | 28 |
| GC5 | 13 Mar 2017 | 24 Mar 2017 | 35 |
| Growing season (2017/2018) | | | |
| Staging grazing | 07 Sept 2017 | 18 Sept 2017 | - |
| GC1 | 04 Dec 2017 | 15 Dec 2017 | 77 |
| GC2 | 03 Jan 2018 | 14 Jan 2018 | 30 |
| GC3 | 05 Feb 2018 | 16 Feb 2018 | 33 |
| GC4 | 12 Mar 2018 | 23 Mar 2018 | 35 |
| GC5 | 23 Apr 2018 | 04 May 2018 | 42 |

[a] The rest period was interrupted and grazing for all treatments re-initiated when one of the treatments reached 95% light interception.

average light interception was 95, 96, and 90% for control, fertilized, and mixed treatments, respectively. Five growth cycles occurred in each rainy season with a rest period (RP) between them varying from 21 to 42 d (Table 1), based upon the seasonal and annual growth variation for time to reach 95% LI. A reserve palisadegrass pasture was used to keep the animals during the rest periods.

## Response variables

**Herbage and canopy characteristics.** All samples were collected from the second paddock on the second grazing day (Fig 2). Before putting the grazing animals on paddocks, the canopy height was estimated using a sward stick at 16 random points. The average canopy height was calculated and two samples representing the average herbage mass were taken using 0.25-m$^2$ quadrats. The herbage biomass at pre-grazing was harvested at a 2 cm stubble height. A subsample taken from the herbage biomass (fertilized and control treatment) was separated into leaf blade (leaf biomass), stem plus sheath (stem biomass), and senescent biomass (brown leaf and stem attached to the plant). When present, the inflorescence was included in the stem fraction. For the mixed treatment, the entire sample was hand separated to measure the proportion of legume and grass (botanical composition). For calculation of herbage biomass at pre-grazing [kg dry matter (DM) ha$^{-1}$], all aboveground biomass (leaf + stem + senescent biomass) was considered; for the mixed treatment, total grass plus legume was considered. These fractions and the remaining non-separated herbage were oven-dried at 55˚C for 72 h to calculate leaf:stem ratio (L:S). Herbage accumulation rate (kg DM ha$^{-1}$ d$^{-1}$) was estimated as the difference between the herbage biomass at pre-grazing of the current growth cycle and the herbage biomass at post-grazing of the previous growth cycle (leaf + stem + senescent biomass was considered in this calculation), divided by the number of rest days between growth cycles (Table 1).

**Nitrogen uptake, fixation, and allocation.** At the beginning of each growth cycle, samples of total aboveground biomass of forage peanut and non-N$_2$-fixing reference plants

(weeds) including billygoat-weed (*Ageratum conyzoides* L.), beggar-ticks (*Bidens pilosa* L.), coat-buttons (*Tridax procumbens* L.), gomphrena-weed (*Gomphrena celosioides* Mart.), cockroach berry (*Solanum capsicoide* All.), and sanguinarea [*Alternanthera ficoidea* (L.) P. Beauv.] were collected from the same area. Different reference plants were collected due to the low number of weeds in the paddock. Oven-dried samples were ground using a Wiley mill (Marconi, Ma680) to pass through a 1-mm stainless steel screen. After samples were roller milled for 16 h [26], total N concentration, and abundance of $^{15}$N (‰) were analyzed using a mass spectrometer (model Delta V Advantage, Thermo Scientific) coupled to an elemental analyzer (model Flash 2000, Thermo Scientific) in the "John Day Stable Isotope Laboratory" (Embrapa Agrobiologia, Seropedica, RJ, Brazil). The total aboveground N of forage peanut (TAGN$_{FP}$, kg N ha$^{-1}$) was calculated by multiplying the N concentration by the herbage accumulation of forage peanut. The percentage of N derived from the atmosphere (%Ndfa) for forage peanut was calculated using the equation described by Shearer and Kohl [27]:

$$\%Ndfa = \left( \frac{\delta^{15}N_{\text{refence}} - \delta^{15}N_{legume}}{\delta^{15}N_{\text{refence}} - B} \right) \times 100$$

where $\delta^{15}N_{\text{reference}}$ is the $\delta^{15}$N value detected in a reference plant (non-N fixing) growing in the same soil at the same growth cycle in the palisadegrass-forage peanut paddock, $\delta^{15}N_{\text{legume}}$ is the $\delta^{15}$N abundance of the forage peanut, and $B$ is the $\delta^{15}$N value of the N in the legume obtained from N$_2$ fixation. We used the $B$ value of −1.35‰ reported by Unkovich [28] for *Arachis hypogea* L. in Brazil. The $\delta^{15}$N from reference plants ranged from 4.4 to 5.9‰ ± 0.6‰, depending upon the sampling date. The total contribution of N from biological N$_2$ fixation (BNF) was calculated by multiplying TAGN$_{FP}$ by the %Ndfa [28].

## Statistical analysis

Data were analyzed using the MIXED procedure of the SAS statistical package (SAS Inst. Inc., Cary, NC, USA). Herbage biomass at pre-grazing, herbage accumulation rate, and morphological components of palisadegrass were analyzed with N management, growth cycle and their interaction as fixed effects, and paddock within N management and year were considered as random effects. Morphological components of forage peanut, TAGN$_{FP}$, %Ndfa, and BNF were analyzed with growth cycle as a fixed effect and paddock within year as a random effect. The Satterthwaite approximation was used to determine the denominator degrees of freedom for the test of fixed effects. The LSMEANS were compared using the PDIFF procedure by Fisher's protected least significant difference (LSD). Differences were considered significant at $P \leq 0.1$. Pearson correlation were carried out to explore the relationship between growth cycle (GC) and rainfall using the CORR procedure of the SAS.

## Results

Nitrogen fertilizer increased the herbage biomass compared with unfertilized palisadegrass and palisadegrass-forage peanut mixture, which did not differ from each other ($P = 0.028$; Fig 3A). The N management × growth cycle interaction effect on the herbage accumulation rate approached significance ($P = 0.106$; Fig 3B), where N fertilization increased the herbage accumulation rate in the GC4 compared to control and mixed treatment. Regarding the GC effect within N management, no difference was found in the herbage accumulation rate among GC for the mixed system. The greatest herbage accumulation rate occurred in the GC2 for the control treatment and GC4 for the fertilized treatment. There was a weak, positive correlation between rainfall and herbage accumulation rate (r = 0.18, $P = 0.085$).

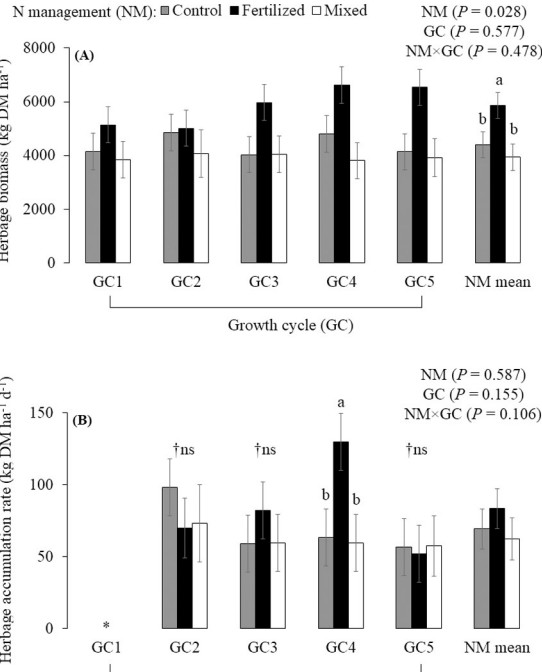

**Fig 3.** Herbage biomass (A) and herbage accumulation rate (B) of palisadegrass under three nitrogen management strategies (NM) for five growth cycles (mean ± SE bars). Control: palisadegrass without N fertilizer or legume; Fertilized: palisadegrass fertilized with 150 kg N ha$^{-1}$ year$^{-1}$; and Mixed: palisadegrass mixed with forage peanut and no N fertilizer. *Growth cycle 1 (GC1) was not evaluated, because herbage was not collected prior to staging in September. †Non-significant. Data are means across 2 years, and 4 replicates (n = 8). Means sharing the same letter are not different, according to the PDIFF procedure by LSD test (P ≤ 0.1).

The mixed treatment herbage mass was separated into palisadegrass and forage peanut to compare the morphological components of palisadegrass among treatments (Table 2). Canopy height, leaf biomass, and stem biomass of palisadegrass were affected by NM and GC, while the L:S ratio was only affected by GC (P ≤ 0.1). Nitrogen fertilization increased canopy height, leaf biomass, and stem biomass of palisadegrass compared with the control and mixed treatments. Canopy height, stem biomass, and L:S ratio of palisadegrass were similar among GC, except in GC5, which had the lowest canopy height. In contrast, the lowest stem biomass, and the greatest L:S ratio occurred at GC1. The greatest leaf biomass of palisadegrass was observed in GC1, while the lowest occurred at GC5. There were positive correlations between rainfall and canopy height (r = 0.23, P = 0.013), leaf biomass (r = 0.31, P < 0.001), and L:S ratio (r = 0.17, P = 0.064).

Senescent biomass was affected by NM × GC interaction (P < 0.001; Fig 4). The senescent biomass varied with N management strategies during the growth cycles. The N fertilization resulted in less senescent biomass in GC1 and GC2 compared with control and mixed treatment, while an increase was observed in GC5. Regarding the GC effect within N management, the lowest senescent biomass was observed in the GC1, GC3, and GC4 for the fertilized, control, and mixed treatment, respectively. The greatest senescent biomass was observed in GC2 and GC4 for the control treatment and in GC5 for fertilized and mixed treatments. There was a negative correlation between rainfall and senescent biomass (-0.28, P = 0.003), i.e., lower rainfall increased the senescent biomass of the palisadegrass.

**Table 2. Morphological components of palisadegrass under three nitrogen management strategies, and forage peanut biomass and nitrogen fixation of forage peanut in the mixed system, for two rainy seasons conducted across five growth cycles.**

| | Nitrogen management strategies (NM) | | | SE | Growth cycle (GC) | | | | | SE | P-value | | |
|---|---|---|---|---|---|---|---|---|---|---|---|---|---|
| | Control | Fertilized | Mixed[a] | | 1 | 2 | 3 | 4 | 5 | | NM | GC | NM×GC |
| *Grass* | | | | | | | | | | | | | |
| CH | 26b | 38a | 27b | 3 | 33a | 31a | 32a | 31a | 25b | 2 | 0.023 | 0.002 | 0.228 |
| Leaf | 1700b | 2490a | 1310c | 130 | 2230a | 1670bc | 1910b | 1970ab | 1390c | 130 | <0.001 | <0.001 | 0.567 |
| Stem | 1330ab | 1970a | 1070b | 280 | 1050b | 1510a | 1450a | 1750a | 1520a | 210 | 0.083 | 0.032 | 0.669 |
| Senescent | 1360 | 1390 | 1220 | 140 | 1010c | 1360b | 1220bc | 1270bc | 1940a | 140 | 0.942 | <0.001 | <0.001 |
| L:S ratio | 1.7 | 1.5 | 1.8 | 0.2 | 2.3a | 1.5b | 1.6b | 1.5b | 1.6b | 0.2 | 0.679 | <0.001 | 0.672 |
| *Legume* | | | | | | | | | | | | | |
| CH | - | - | 7.4 | 1.3 | 8.7 | 7.8 | 8.3 | 8.0 | 4.2 | 1.3 | - | 0.104 | - |
| FP | - | - | 239 | 126 | 263 | 321 | 304 | 249 | 60 | 126 | - | 0.143 | - |
| L:S ratio | - | - | 1.1 | 0.2 | 1.4a | 1.2abc | 1.3ab | 0.9bc | 0.8c | 0.2 | - | 0.092 | - |
| $TAGN_{FP}$ | - | - | 3.1 | 2.5 | 6.8a | 1.9b | 4.1ab | 2.1b | 0.6b | 2.5 | - | 0.099 | - |
| %Ndfa | - | - | 82 | 2 | 79c | 81bc | 82ab | 82b | 85a | 2 | - | 0.029 | - |
| BNF | - | - | 2.5 | 2.0 | 5.5a | 1.5ab | 3.2a | 1.7ab | 0.5b | 2.0 | - | 0.010 | - |

[a] Mixed treatment was separated into palisadegrass and forage peanut data.

CH, canopy height (cm); leaf biomass (kg DM ha$^{-1}$); stem biomass (kg DM ha$^{-1}$); senescent biomass (kg DM ha$^{-1}$); L:S, leaf: stem ratio; FP, forage peanut biomass (kg DM ha$^{-1}$); $TAGN_{FP}$, total aboveground N of forage peanut (kg N ha$^{-1}$ cycle$^{-1}$); %Ndfa, percentage of nitrogen derived from the atmosphere by forage peanut; BNF, biological N$_2$ fixation (kg N ha$^{-1}$ cycle$^{-1}$); SE, standard error.

Forage peanut canopy height, L:S ratio, total aboveground N ($TAGN_{FP}$), and BNF showed a reduction, mainly for GC5 ($P \leq 0.1$), while percent nitrogen derived from the atmosphere (%Ndfa) increased in the same cycle. There was a positive correlation between rainfall and canopy height (r = 0.41, $P = 0.017$) and L:S ratio (r = 0.54, $P = 0.001$). The botanical composition of forage peanut and palisadegrass in the mixed treatment differed among GCs ($P < 0.001$). However, these proportions did not change among GC1 to GC4 (ranging between 7 to 9% of the legume in the botanical composition), except in GC5, which had the least forage peanut proportion (2% of the botanical composition).

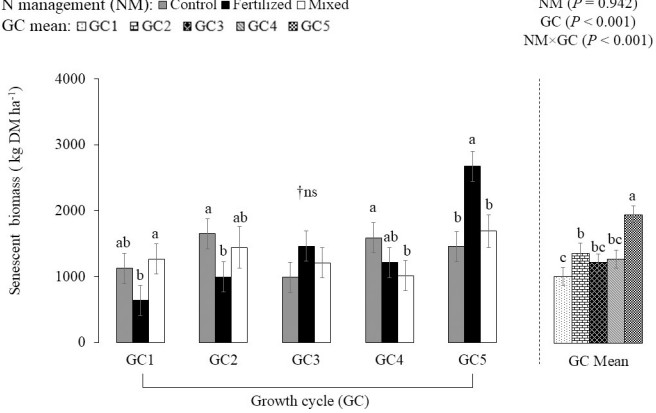

**Fig 4. Nitrogen management strategies (NM) × growth cycle (GC) interaction effect on senescent biomass of palisadegrass (mean ± SE bars).** Control: palisadegrass without N fertilizer or legume; Fertilized: palisadegrass fertilized with 150 kg N ha$^{-1}$ year$^{-1}$; and Mixed: palisadegrass mixed with forage peanut and no N fertilizer. †Non-significant. Data are means across 2 years, and 4 replicates (n = 8). Means sharing the same letter within GC are not different, according to the PDIFF procedure by LSD test ($P \leq 0.1$).

## Discussion

The goal of our study was to test whether there was a meaningful and measurable contribution of N from forage peanut, even at minimal participation in the mixed system. Nitrogen is generally the most limiting nutrient for plant growth. It enhances tillering [29], leaf number and/or leaf elongation [30] and is a major component of the chlorophyll molecule [31]. Results of studies that accounted for the BNF capacity of forage peanut to fix the nitrogen derived from atmospheric (%Ndfa) were consistent with our findings. Assessments developed in the Southeastern United States [12, 32] and Northeast and Southeastern Brazil [14, 33] have shown that the %Ndfa for rhizoma peanut and forage peanut can vary between 60 to 96%, regardless of whether it was grown as monocultures or mixtures. However, BNF-N by the forage peanut in our study was insufficient to promote palisadegrass growth beyond that found under unfertilized palisadegrass management.

In the present study, the total amount of BNF was 15.5 kg N ha$^{-1}$ cumulated across five growth cycles. The weather and soil conditions are considered key drivers that regulate the BNF, mainly by affecting bacterial activity [34]. However, no correlation between rainfall and BNF was found in the present study (r = 0.26, $P$ = 0.128). Relatively low BNF was most likely due to the reduced participation of forage peanut in the botanical composition ($< 10\%$) of the mixed system. Legume proportions between 20–45% in the herbage biomass have been considered sufficient to supply N to the agroecosystem [19] and to fix between 50–100 kg N ha$^{-1}$ yr$^{-1}$ [35].

In the current study, the N supply from synthetic fertilizer enhanced herbage biomass by 33 and 49% and leaf biomass by 46 and 90% of palisadegrass compared with unfertilized palisadegrass and palisadegrass-forage peanut mixture, respectively (Fig 3; Table 2). Nitrogen fertilization increases cell division, increasing leaf elongation [36], as well as tiller growth and turnover. These factors can vary with temperature and rainfall [37]. In addition, the N fertilized treatment always reached the 95% light interception threshold before the other treatments. Therefore, the low N treatments (grass or mixture) had greater grazing pressure placed upon them, which may have weakened stands.

In the present study, the rainfall was positively correlated with canopy height (r = 0.23, $P$ = 0.013), leaf biomass (r = 0.31, $P$ < 0.001), and L:S ratio (r = 0.17, $P$ = 0.064) of palisadegrass. Stem elongation is not often desirable for grazing animal nutrition. Increasing stem growth can reduce the leaf proportion [38]. Stem elongation increases the canopy height, which will affect the ingestive animal behavior, such as reducing the number of tongue sweeps, bite volume, and bite weight [38, 39]. Stem acts as a leaf support structure with lignified secondary cell and other less digestible components [40, 41], providing stiffness to the stem. Combined, these traits may reduce animal performance [42].

Nitrogen fertilization combined with favorable weather conditions promoted greater green biomass (leaf and stem) and reduced the senescent biomass, while the N fertilization and drought conditions of the transition period between rainy to dry season may have increased senescent biomass. A negative correlation occurred between senescent biomass and rainfall (-0.28, $P$ = 0.003). There was more senescent biomass in the N fertilized treatment because there was more living biomass (leaf + stem). However, when senescent biomass was compared to the living biomass as a ratio, the N fertilizer treatment had a much lower proportion of senescent to living biomass, representing only 24% of the total herbage biomass. In contrast, in the unfertilized and mixed treatments, the senescent biomass represented 31 and 34% of the total herbage mass, respectively. The low N fertilization may have hastened the plant growth cycle which promoted greater senescence.

Under regular weather conditions (rainfall, temperature, and photoperiod), warm-season $C_4$ grasses are typically more competitive than legumes, shifting the balance in mixed swards towards the grass [14]. This competition can reduce forage peanut persistence in mixed systems. Grass establishment, growth, and time to recover after grazing tend to be faster with $C_4$ grasses than with legumes [10]. We observed a reduction in the presence of forage peanut, probably due to the dry conditions during the transition period from rainy to dry season. During the dry season, forage peanut typically exhibits leaf loss [43]. In Northwest Brazil, drought stress reduced forage peanut biomass in a mixed system [17]. On the other hand, in Northeast Brazil, there was no negative impact of the dry season on forage peanut growth [44]. The authors suggested positive legume performance due to the favorable weather in the study region, even during the dry season. Thus, differences among Brazilian regions, weather conditions, and latitude play an important role in the forage peanut growth, mainly during the dry season. With the onset of the rainy season, forage peanut initiates growth. The transition period between dry to rainy season is critical for forage peanut recovery due to the much lower grass competition and greater patches of bare ground that allow germination of the soil seed bank after the first rains [17, 45]. Even so, recovery occurs slowly [46].

Keeping the light interception between 90 and 95% with canopy height ranging from 26 to 32 cm for palisadegrass-forage peanut mixtures was considered the best grazing management to provide greater herbage accumulation rate, especially leaves [25]. However, in our study with a minimal proportion of forage peanut, it may be necessary to keep the LI less than 90% in order to reduce grass competition with the forage peanut [47]. It has been reported that a sward canopy height of 20 cm for mixed plantings, mainly in the first years after seeding without compromising the grass growth [15, 48]. Increased grass canopy height can also stimulate forage peanut vertical growth, thereby, reducing the number of stolons and generally decreasing legume proportion in the mixed system.

Further studies are required to evaluate whether N fertilization applied at key periods (when conditions are more favorable to forage peanut growth than grass growth) could hasten the legume proportion, at least during the initial years. These factors require further research in pursuit of developing sustainable mixed systems that provide quality feed to grazing animals, provide cost reduction to the farmer and lessen the environmental impacts of synthetic N fertilizers.

## Conclusions

A small amount of forage peanut in a pasture with palisadegrass gave similar results to unfertilized palisadegrass for most response variables. Botanical composition fluctuated according to the growth cycle, with forage peanut disappearance at the beginning of the transition from rainy to dry season. The total input form biological $N_2$ fixation decreased with the reduction of forage peanut biomass, which was affected by periodic dry weather conditions during the season. The forage peanut inclusion was not enough to increase pasture productivity above the non-fertilized palisadegrass monoculture. The N fertilized palisadegrass monoculture had greater herbage biomass and leaf: stem ratio. It appears that the forage peanut was less tolerant than palisadegrass to dry conditions as was noted with the transition to the dry season.

## Supporting information

**S1 Data.**
(XLSX)

## Acknowledgments

The authors thank the members of UnespFOR (Brazilian forage Team) for the contributions during the field trial setup.

## Author Contributions

**Conceptualization:** Vanessa Z. Longhini, Abmael S. Cardoso, Andressa S. Berça, Robert M. Boddey.

**Data curation:** Vanessa Z. Longhini, Abmael S. Cardoso, Andressa S. Berça, José C. B. Dubeux, Jr.

**Formal analysis:** Vanessa Z. Longhini, Abmael S. Cardoso, Andressa S. Berça, José C. B. Dubeux, Jr., Ana C. Ruggieri.

**Funding acquisition:** Vanessa Z. Longhini, Abmael S. Cardoso, Robert M. Boddey, Ricardo A. Reis.

**Investigation:** Vanessa Z. Longhini, Abmael S. Cardoso, Andressa S. Berça.

**Methodology:** Vanessa Z. Longhini, Abmael S. Cardoso, Andressa S. Berça, Robert M. Boddey, Ricardo A. Reis, José C. B. Dubeux, Jr., Ana C. Ruggieri.

**Project administration:** Vanessa Z. Longhini, Abmael S. Cardoso, Andressa S. Berça, Robert M. Boddey, Ana C. Ruggieri.

**Resources:** Vanessa Z. Longhini, Abmael S. Cardoso, Robert M. Boddey, Ricardo A. Reis, Ana C. Ruggieri.

**Supervision:** Vanessa Z. Longhini, Abmael S. Cardoso, Andressa S. Berça, Robert M. Boddey, Ana C. Ruggieri.

**Visualization:** Ana C. Ruggieri.

**Writing – original draft:** Vanessa Z. Longhini, Abmael S. Cardoso, Andressa S. Berça, Robert M. Boddey, Ricardo A. Reis, José C. B. Dubeux, Jr., Ana C. Ruggieri.

**Writing – review & editing:** Vanessa Z. Longhini, Abmael S. Cardoso, Andressa S. Berça, Robert M. Boddey, Ricardo A. Reis, José C. B. Dubeux, Jr., Ana C. Ruggieri.

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
