## [Decision Letter · Decision Letter 0]

27 Aug 2020

PONE-D-20-17268

Impact of N fertilization or the presence of forage peanut on the herbage structure of palisadegrass under grazing

PLOS ONE

Dear Dr. Longhini,

Thank you for submitting your manuscript to PLOS ONE. After careful consideration, we feel that it has merit but does not fully meet PLOS ONE’s publication criteria as it currently stands. Therefore, we invite you to submit a revised version of the manuscript that addresses the points raised during the review process.

A major problem with the paper – as highlighted by both reviewers – is that the authors goal is to compare the legume-grass mixture with the grass monoculture, but they failed to achieve an adequate amount of legume in the mixture to provide a legitimate test of the mixture's potential. Thus, although they gathered a lot of interesting data, they did not adequately test their hypotheses because their legume-grass mixture had much too little legume to provide a valid test. It may be possible to develop an interesting study even though the pintoi did not perform as expected. As reviewers suggest, the use the weather and growth cycle data to find correlations (if they exist) and thus an explanation of results.

There is also a substantial lack of organization, excess detail and insufficient attention to testing of hypotheses that they can achieve, given what the study actually tested. I agree with reviewer two that the results and discussion should be separated, and the need to reduce by about 50% including all redundant measurements, and figures that are redundant with tables. 

So this is a really major revision needed, in terms of reanalysis and rewriting - just so this is a clear recommendation and I hope that this is undertaken as the work is important, just challenging given that the original hypotheses were not possible to test due to  establishment issues.

We look forward to receiving your revised manuscript.

Kind regards,

Sieglinde S. Snapp

Academic Editor

PLOS ONE

Additional Editor Comments:

A major problem with the paper – as highlighted by both reviewers – is that the authors goal is to compare the legume-grass mixture with the grass monoculture, but they failed to achieve an adequate amount of legume in the mixture to provide a legitimate test of the mixture's potential. Thus, although they gathered a lot of interesting data, they did not adequately test their hypotheses because their legume-grass mixture had much too little legume to provide a valid test. It may be possible to develop an interesting study even though the pintoi did not perform as expected. As reviewers suggest, the use the weather and growth cycle data to find correlations (if they exist) and thus an explanation of results.

There is also a substantial lack of organization, excess detail and insufficient attention to testing of hypotheses that they can achieve, given what the study actually tested. I agree with reviewer two that the results and discussion should be separated, and the need to reduce by about 50% including all redundant measurements, and figures that are redundant with tables.

Journal Requirements:

Reviewers' comments:

Reviewer's Responses to Questions

**Comments to the Author**

1. Is the manuscript technically sound, and do the data support the conclusions?

Reviewer #1: Yes

Reviewer #2: Yes

2. Has the statistical analysis been performed appropriately and rigorously? 

Reviewer #1: Yes

Reviewer #2: Yes

3. Have the authors made all data underlying the findings in their manuscript fully available?

Reviewer #1: Yes

Reviewer #2: Yes

4. Is the manuscript presented in an intelligible fashion and written in standard English?

Reviewer #1: Yes

Reviewer #2: Yes

5. Review Comments to the Author

Reviewer #1: The paper is generally well written. In the attached pdf, there are comment boxes suggesting revisions or asking questions. The biggest issue the authors are dealing with is that the establishment of pintoi was not satisfactory. This impacts any conclusions about the three treatments, because you are comparing the two strong grass monocultures against a mixture that does not reflect the true potential of the mixture.

The authors try to get around this by saying in one case that they are describing what happens during the establishment period, but in another case they say they are addressing whether there is meaningful and measurable contribution from peanut even at a very low participation. I encourage them to focus the paper on one of these two approaches, not both of them. I think the latter is most plausible. The problem with saying they are looking at what happens during establishment is that the planting occurred in 2014 and the experiment did not start until 2016. Yes, I very well know that Arachis establishes slowly, but to suggest that Years 3 and 4 after planting are the establishment period is a rather large stretch. I suggest that instead they focus on whether there is value even of a small proportion of peanut in the association and do that intentionally and consistently throughout the paper so that the theme is well developed and serves as a point of focus.

Reviewer #2: The authors report on the response of palisade pasture with no N inputs, mineral N, or Pintoi peanut (after 2-year establishment phase). The legume did not benefit sward productivity, which was assumed to be due to inadequate establishment length and periodic dry periods across several grazing cycles. Negative results in research are useful if well-delivered and if it addresses important topics. Since mineral N replacement is of great interest in food production, this topic has value. However, there are some major challenges the authors need to address. Many were included in the attached Word document (see attachment). Other notable points:

- Experimental design as a Completely Randomized Design. Is that the case? Typically, these experiments are Randomized, Complete Block Design. It might be beneficial to include a figure with the experimental design illustrated.

- The manuscript was exceptionally long and many of the measurements were variations of key measurements, which made several parts of the Results and Discussion section seem redundant. I think the Conclusions section exemplified the simplicity of the results and the authors should use that as a guide on how to address their results in the main body of the document. Biomass was measured, leaf biomass, stem biomass, green herbage, dead herbage (not sure how that is defined), and daily growth rates. The results were clear but the authors greatly complicate the results with long discussion on the each of these (and several other) measurements that all lead to the same conclusion (the pintoi did not contribute to productivity as much as N fertilizer). Not much of a mystery there! I suggest dropping much of the redundant measurements, figures that are redundant with tables and focus on the key points. The pintoi did not perform as expected and to use the weather and growth cycle data to find correlations (if they exist) and explanation of results. This manuscript can easily be reduce by 50% without impacting the message. Just because you measured lots of things does not mean they all need to be reported on! Pick the key points and focus on them. Keep the reader interested!

- To help reduce redundancy and be more textually efficient, it may help to structure this manuscript with separate Results and Discussion sections rather than combining them as Results and Discussion. Then the Discussion section can flow freely without stepping through each measurement one by one and repeating the same cause-effect for every dependent variable.

- Paragraphs are not consistently indented (run-on paragraphs).

- Was not sure how the 3 N fertilizer split applications factored into grass response (growth cycles) in terms of biomass, as some cycles had greater biomass than others. Seems that would impact biomass differently over the season.

- Make sure any abbreviations used in tables and figures are written out (in title, footnote, caption, as appropriate).

6. PLOS authors have the option to publish the peer review history of their article (what does this mean?). If published, this will include your full peer review and any attached files.

Reviewer #1: No

Reviewer #2: No

---

## [Author Response · Author response to Decision Letter 0]

19 Nov 2020

Jaboticabal, Sao Paulo State, Brazil, November 18th, 2020.

Dear Plos One Editor,

Subject: Submission of a revised manuscript PONE-D-20-17268 entitled “Could forage peanut in low proportion replace N fertilizer in livestock systems?”.

Thank you for considering our manuscript “Could forage peanut in low proportion replace N fertilizer in livestock systems?” for publication. We carefully addressed each of the reviewers’ comments, and we hope that the changes will meet your expectations. Our answers are listed below (in red) and a version of the revised manuscript with all the modifications (i.e., corrections and additional information inserted) highlighted in “track-change mode”. We would like to thank you and the reviewers for the excellent review, which certainly improved the manuscript quality. 

Thank you for the suggestions and possible appreciation of our work for publication in this renowned journal. The authors are available for other modifications that may improve the quality of the manuscript, as well as additional clarifications.

Sincerely,

Vanessa Zirondi Longhini, Ph.D.

Department of Animal Science

São Paulo State University (UNESP-FCAV)

Jaboticabal (SP), Zip 14884-900, Brazil

Comments to Author

Additional Editor Comments:

1. Thank you for submitting your manuscript to PLOS ONE. After careful consideration, we feel that it has merit but does not fully meet PLOS ONE’s publication criteria as it currently stands. Therefore, we invite you to submit a revised version of the manuscript that addresses the points raised during the review process. A major problem with the paper – as highlighted by both reviewers – is that the authors goal is to compare the legume-grass mixture with the grass monoculture, but they failed to achieve an adequate amount of legume in the mixture to provide a legitimate test of the mixture's potential. Thus, although they gathered a lot of interesting data, they did not adequately test their hypotheses because their legume-grass mixture had much too little legume to provide a valid test. It may be possible to develop an interesting study even though the pintoi did not perform as expected. 

Authors Response: Many thanks for considering a new version of the manuscript. After careful revision, we changed the focus of the manuscript. Please, see the hypothesis in the introduction section: “We hypothesized that the forage peanut mixed with non-N fertilized grass would benefit the herbage responses through BNF, compared with unfertilized grass in a monoculture. However, the proportion of forage peanut in the sward in the present study was low. Thus, we decided to test whether there is a meaningful and measurable contribution of the N from forage peanut even at minimal participation and whether this contribution could be close to those systems using synthetic N fertilizer. Therefore, this study evaluated the performance of palisadegrass under three nitrogen management strategies (unfertilized, N fertilized, or mixed with the legume forage peanut), aiming to identify how different N inputs may affect grass responses”. We rewrote the results and discussion in separate sections.

2. As reviewers suggest, the use the weather and growth cycle data to find correlations (if they exist) and thus an explanation of results. 

Authors Response: Thank you for this suggestion. We checked the correlations, and they were included in the manuscript.

3. There is also a substantial lack of organization, excess detail and insufficient attention to testing of hypotheses that they can achieve, given what the study actually tested. I agree with reviewer two that the results and discussion should be separated, and the need to reduce by about 50% including all redundant measurements, and figures that are redundant with tables. So this is a really major revision needed, in terms of reanalysis and rewriting - just so this is a clear recommendation and I hope that this is undertaken as the work is important, just challenging given that the original hypotheses were not possible to test due to establishment issues.

Authors Response: We fully agree with this critical and useful comments. Please, note that we split the results and discussion section, removed redundant measurements and figures, which was possible to reduce the manuscript.

Reviewers' comments to author

Reviewer 1

The paper is generally well written. In the attached pdf, there are comment boxes suggesting revisions or asking questions. The biggest issue the authors are dealing with is that the establishment of pintoi was not satisfactory. This impacts any conclusions about the three treatments, because you are comparing the two strong grass monocultures against a mixture that does not reflect the true potential of the mixture. The authors try to get around this by saying in one case that they are describing what happens during the establishment period, but in another case they say they are addressing whether there is meaningful and measurable contribution from peanut even at a very low participation. I encourage them to focus the paper on one of these two approaches, not both of them. I think the latter is most plausible. The problem with saying they are looking at what happens during establishment is that the planting occurred in 2014 and the experiment did not start until 2016. Yes, I very well know that Arachis establishes slowly, but to suggest that Years 3 and 4 after planting are the establishment period is a rather large stretch. I suggest that instead they focus on whether there is value even of a small proportion of peanut in the association and do that intentionally and consistently throughout the paper so that the theme is well developed and serves as a point of focus.

Authors Response: Thank you very much for your effort in revising our manuscript. All suggestion made in the pdf file was included in the new version. We agree with you on all points raised during the revision process. We are now focusing on the small contribution of Arachis pintoi to the system instead of the legume establishment process. Please, note that some sentences and references were removed during the review process to reduce redundancy and be more textually efficient, as suggested by the editor and reviewer 2.

1. This is not true everywhere all the time, so I suggest "The current relationship between beef price and fertilizer in Brazil does not".

Authors Response: We appreciate this comment. The sentence was corrected according to the suggestion (line 21).

2. vegetation types implies species or growth habit, suggest using "sward management strategies" as you do later, or perhaps "pasture species composition".

Authors Response: Thank you for the suggestion. We changed to “pasture species composition” as suggested (line 24).

3. I prefer "season" because trial implies a different experiment, and you simply repeated the same experiment in two year, at least that is what I understand.

Authors Response: We agree with the reviewer. We changed to “season” as suggested (line 196).

4. In-stead of "Herbage structures", I suggest "Herbage and canopy characteristics".

Authors Response: Thank you for the suggestion. We have made changes as suggested (line 207).

5. I am not sure what you mean by this sentence. If height is not consistent with LI, then how can you use the same height for comparisons? Expand what you are trying to say here so that it is more clear. What do you mean by lower cost? Easier and cheaper to measure height than LI???

Authors Response: We agree with your comment. The sentence was confusing. After careful revision, we removed the sentence from the new version.

6. suggest "herbage accumulation rates in monoculture or mixed pastures". The word especially does not seem to fit when you are discussing more than one item.

Authors Response: The sentence was removed in the new version.

7. occurred due to drought conditions.

Authors Response: Thank you for the suggestion. We rephrased the sentence in the new version: “Nitrogen fertilization combined with favorable weather conditions promoted greater green biomass (leaf and stem) and reduced the senescent biomass, while the N fertilization and drought conditions of the transition period between rainy to dry season may have increased the senescent biomass”.

8. resulting in lesser green herbage mass (leaf and stem) and greater dead mass in GC5.

Authors Response: Thank you for the suggestion. We rephrase the sentence in the new version (Please, see the last answer).

9. interception threshold before the other treatments. This may have resulted in competition for light among tillers, and the plant elongates stem under these conditions in search of light for the younger leaves.

Authors Response: Many thanks for your suggestion. We have made changes as suggested (lines 379 and 383-384).

10. season, tillers may have increased in number or weight [34].

Authors Response: Removed from the new version.

11. It isn't totally clear to me how you planted in 2014 and started the study in 2016, yet you describe the period of study as the establishment period. It seems that you simply had an inferior stand of peanut, and it was never going to get better.

Authors Response: You are correct. We are now focusing on the forage peanut contribution to the system, even at minimal participation. This sentence was removed from the new version.

12. Keep in mind that the mixed treatment had only 90% LI on average, right? So, you would need to reduce this number to considerably less than 90, would you not?

Authors Response: We are grateful for this comment. We agree with you. In the mixed system the grazing started at 90% LI, therefore, we changed to “less than 90% of light interception” as suggested (line 805). 

Reviewer 2

The authors report on the response of palisade pasture with no N inputs, mineral N, or Pintoi peanut (after 2-year establishment phase). The legume did not benefit sward productivity, which was assumed to be due to inadequate establishment length and periodic dry periods across several grazing cycles. Negative results in research are useful if well-delivered and if it addresses important topics. Since mineral N replacement is of great interest in food production, this topic has value. However, there are some major challenges the authors need to address. Many were included in the attached Word document (see attachment). 

Authors Response: Thank you very much for all the valuable advice and points raised during the revision process. We hope that all changes will meet your expectations. We included the suggestions made in both files.

Other notable points:

- Experimental design as a Completely Randomized Design. Is that the case? Typically, these experiments are Randomized, Complete Block Design. It might be beneficial to include a figure with the experimental design illustrated.

Authors Response: Yes, the experimental design was in a completely randomized design. We included a figure with the experimental design (Figure 2).

- The manuscript was exceptionally long and many of the measurements were variations of key measurements, which made several parts of the Results and Discussion section seem redundant. I think the Conclusions section exemplified the simplicity of the results and the authors should use that as a guide on how to address their results in the main body of the document. Biomass was measured, leaf biomass, stem biomass, green herbage, dead herbage (not sure how that is defined), and daily growth rates. The results were clear but the authors greatly complicate the results with long discussion on the each of these (and several other) measurements that all lead to the same conclusion (the pintoi did not contribute to productivity as much as N fertilizer). Not much of a mystery there! I suggest dropping much of the redundant measurements, figures that are redundant with tables and focus on the key points. The pintoi did not perform as expected and to use the weather and growth cycle data to find correlations (if they exist) and explanation of results. This manuscript can easily be reduce by 50% without impacting the message. Just because you measured lots of things does not mean they all need to be reported on! Pick the key points and focus on them. Keep the reader interested!

Authors Response: We agree with you. We removed some redundant data and separated the results from the discussion section to reduce the manuscript. We are now focusing on the small contribution of Arachis pintoi to the system instead of the legume establishment process. Correlations were tested and were included as an explanation of the results. The term “dead herbage” was changed to “senescent biomass” and was defined in the Materials and methods section as “senescent biomass (brown leaf and stem attached to the plant)”. We hope that all changes will meet your expectations.

- To help reduce redundancy and be more textually efficient, it may help to structure this manuscript with separate Results and Discussion sections rather than combining them as Results and Discussion. Then the Discussion section can flow freely without stepping through each measurement one by one and repeating the same cause-effect for every dependent variable.

Authors Response: Many thanks for your suggestion. Please, note that we separated the Results and Discussion section, in this way, the manuscript was reduced without changing the message.

- Paragraphs are not consistently indented (run-on paragraphs).

Authors Response: Thank you for catching it. We revised according to the pdf in the submission guidelines.

- Was not sure how the 3 N fertilizer split applications factored into grass response (growth cycles) in terms of biomass, as some cycles had greater biomass than others. Seems that would impact biomass differently over the season.

Authors Response: We split the application to reduce possible N losses through volatilization, leaching, or runoff. However, some losses may have occurred in the first growth cycles due to high rainfall after urea application. However, the biomass in the fertilizer treatment was greater than in the mixed and control treatment. To improve the results visualization, we change the table for the figure (Fig 3). Please note that the fertilized treatment had the same pattern across all growth cycles.

- Make sure any abbreviations used in tables and figures are written out (in title, footnote, caption, as appropriate).

Authors Response: Checked.

1. Do not understand the notations, in terms of the different years for each grazing evaluation. Might be easier to follow if presented as a table, with a column for each grazing season.

Authors Response: Many thanks for your suggestion. We included the date information in Table 1.

2. This paragraph is probably better suited for the Discussion section.

Authors Response: We appreciate this suggestion. We skip the sentence to the Discussion section (lines 422-424). 

3. Include the year.

Authors Response: We included the year in the revised manuscript.

4. Were the GC equally separated in time? If not, then repeated measures cannot be used. Difficult to determine by the way the sampling dates are presented. The last sampling date seems the most suspicious.

Authors Response: The growth cycle was not equally separated because we used the intermittent stocking method, i.e., GC with irregular intervals (Allen et al., 2011). The rest period was interrupted only when the first treatment was achieved at 95% of light interception. We included Table 1 with the date of the growth cycle. Therefore, with your affirmation, we reanalyzed the statistics removing the repeated measures.

5. Why is forage peanut addressed separately? It seems that treatment (regardless of which one) is addressed as a single factor. Not sure I understand why you separate the different treatments (grass ANOVA vs peanut ANOVA).

Authors Response: We separated the forage peanut results from the mixed treatment to evaluate the palisadegrass performance among nitrogen management strategies and the forage peanut patterns among the growth cycle. Thus, forage peanut was statistically compared among GC (fixed effect) without the other treatments.

6. You should not imply that there were significant differences when there was not by talking in terms of percent differences. The PHM was not different between control and fertilized grass.

Authors Response: Corrected according to the new statistical analyses (P ≤ 0.1). Please, see lines 373-376.

7. If you had legume growth in the mixture but the sward yield was unchanged, then it must have reduced the grass growth! What you might say instead is that inclusion of the legume did not impact sward productivity compared to grass monoculture.

Authors Response: Removed from the new version.

8. Abbreviations need to be defined.

Authors Response: Revised.

9. If true then that would be a SM x GC interaction, which you did not have. All you can say is that there was a GC effect and explain what it was. GC response was independent of SM.

Authors Response: Revised according to the new statistical analyses (P ≤ 0.1). Please, see lines 285-290.

10. Unclear how this statement explains the reported response. Either explain more thoroughly or omit the sentence.

Authors Response: The sentence was removed.

11. I am certain there are water response literature available to support your supposition about Arachis pintoi response to drought or dry soils. We know this to be the case in the US.

Authors Response: Thank you for this suggestion. We included two references to support this supposition: “In Northwest Brazil, [17] reported that the drought stress reduced the forage peanut biomass in the mixed system during the dry season. On the other hand, in Northeast Brazil, [46] did not find a negative impact of the dry season on the forage peanut growth. The authors suggested positive legume performance due to the favorable climatic weather in the study region, even during the dry season. Thus, differences among Brazilian regions, weather conditions, and latitude play an important role in the forage peanut growth, mainly in the dry season” (lines 411-417). 

12. I think this is more a factor of large variability in your data set. Since the yields were functions of the growth rate and they were significantly different, then it would reason that the growth rate would also differ. You have SM x GC interaction (p=0.1) for HAR. For field studies, sometimes using 0.1 as your limit is acceptable.

Authors Response: Many thanks for your suggestion. We are now considering the P ≤ 0.1 for all variables, including the correlations. Please, see lines 285-292.

13. Not easy to see how Fig 1 (given in dates) relates to the growth response (given as cycle numbers). You should consider how to compare them to better support your argument.

Authors Response: Thank you. We changed the figure grouping the date by growth cycle to support the data. We also performed a correlation between rainfall and growth cycle to explain the results.

14. What time of year. It’s a real challenge to keep scrolling back to determine when each cycle was compared to the season/date.

Authors Response: The greater L:S ratio occurred in the first growth cycle (November 2016 and December 2017) at the onset of the rainy season in Brazil. The growth cycle dates were detailed in Table 1.

15. Basically, the extra N contributed to greater stem growth. That seems reasonable. It seems that extra N would either have no effect or decrease L:S. As part of a sward, there is no benefit for a plant to creates relatively more leaf mass, when it cannot intercept anymore light than what it already has. A quick check, led to this example. I’m sure you can find more relevant references, as well: Grass and Forage Science 54(3):255 – 266 DOI: 10.1046/j.1365-2494.1999.00178.x

Authors Response: Thank you for this suggestion. More references was included about the extra N: “The extra N may have contributed to competition for light among tillers, and the plant elongates the stem under these conditions in search of the light for the younger leaves [30]. As part of a canopy sward, there is no benefit for a plant to create relatively more leaf biomass when it cannot intercept anymore light than it already has [38], which may explain the greatest stem proportion and canopy height in the N fertilized treatment without effect on the L:S ratio among N management strategies. In the present study, the rainfall was positively correlated with canopy height (r = 0.23, P = 0.013), leaf biomass (r = 0.31, P < 0.001), and L:S ratio (r = 0.17, P = 0.064) of palisadegrass. Stem elongation is not desirable for grazing animal nutrition, basically for three reasons. First, increasing the stem can reduce the leaf proportion [39]; second, the stem elongation increases the canopy height, which will affect the ingestive animal behavior, such as reducing the number of tongue sweeps, bite volume, and bite weight [39, 40]; and third, the stem acts as a support structure of leaves, which has secondary cell walls, such as lignin, which are undigestible components [41, 42]. These factors can limit the selection of leaf and, consequently, reduces animal gain [43]” (lines 383-396).

16. Why is leaf:stem discussed under “herbage structures” instead of morphological components?

Authors Response: Thank you for catching it. We included the leaf:stem on the morphological components.

17. Leaf:stem ratio is derived from these measurements. Therefore, leaf:stem ratio should be placed in this section.

Authors Response: You are correct. We included the L:S ratio in the morphological components. 

18. I do not understand this argument. I suspect that initiating grazing at 95% light interception was not well maintained rather than canopy heights differing at exactly 95% light interception. I do not think you could have managed the systems that accurately, based upon the information provided. An alternative interpretation is does this mean that canopy height was measured at less than 95% light interception for the unfertilized grass treatment? My interpretation of the data is that the low N treatments (grass or mixture) did not grow as much over the same amount of time as the N fertilized grass. Shows up in biomass and related measurements. No surprise…

Authors Response: Thank you for the critical and useful comments. After careful revision, we removed the sentence and included the suggestion (line 380-382).

19. Abbreviations need to be written out the first time they are used in the Materials and Methods or Results and Discussion section, regardless if they were used in the Introduction.

Authors Response: We appreciate this comment. We checked the abbreviations in the whole manuscript.

20. You did not have grazing frequency comparisons, but rather repeated grazing (cycles). There is nothing presented that suggests that you were testing different frequencies.

Authors Response: We appreciate this comment. We removed the sentence to fix it.

21. This is a key point and brings to question why this study was attempted if the peanut was not established. The message seems to be that in terms of yield, the peanut provides no benefit. IF you had addressed forage quality, animal gain, etc., then perhaps you would have found benefit of the mixed sward. How many years does it take to declare the peanut fully established in a grass sward?

Authors Response: Many thanks for this useful comment. We included a sentence to bring this key point: “Pereira [46] reported that 20.5 to 23.8% of forage peanut mixed with palisadegrass resulted in a liveweight gain per ha 29% greater than in N fertilized system (dose of 120 kg N ha-1). Therefore, if our study had addressed nutritive value and animal performance, perhaps we would have found the mixed system benefits even at minimal legume participation” (lines 427-431). Based on Tamele et al. (2018), in similar environmental conditions of this study, after three years of establishment with the canopy height of palisadegrass at 20 cm, it was possible to guarantee a successful forage peanut establishment with 19 to 38% of the legume in the botanical composition. However, we are now focusing on the small contribution of Arachis pintoi to the system instead of the legume establishment process.

Tamele OH, Sá OAAL, Bernardes TF, Lara MAS, Casagrande DR. Optimal defoliation management of brachiaria grass–forage peanut for balanced pasture establishment. Grass Forage Sci. 2018; 73:522–531. doi: http://dx.doi.org/10.1111/gfs.12332

22. This is much greater than your reported values, so what can the reader conclude about your system?

Authors Response: You are right. However, we rewrote the sentence to support the last answer (Please, see lines 427-429).

23. Maybe true for grasses, particularly those that lose quality, with time between grazing events, but do you expect to the same extent (or at all) with the peanut species? I doubt it. You can use this argument for the grass discussion section.

Authors Response: We appreciate this comment, you are correct. We moved the sentence to the grass discussion section (lines 392-393 and 395-396). 

24. Across all conditions?

Authors Response: Many thanks for your comment. Please, note that we included the sentence: “Under regular weather conditions (rainfall, temperature, and photoperiod), warm-season C4 grasses are typically more competitive than legumes, shifting the balance in mixed swards towards the grass [14]” (lines 405-407). 

25. Rhizoma peanut is not “seeded”, thus the name “rhizoma” peanut (Arachis glabrata). It is sprigged.

Authors Response: Thank you for catching it. However, the sentence was removed in the revision.

26. I do not understand what you are trying to say (in the dung?). You have the biomass values. You can simply compare proportions based on biomass.

Authors Response: We were trying to say that although the legume proportion in the botanical composition was low, the animals selected the forage peanut to compose their diet, and this affirmation we could perform by the isotopic 13C in the dung, which was numerically greater than in the botanical composition. However, we removed the sentence to avoid confusion. 

27. Are the last two sentences addressing Reference 45 or your study? This is unclear and the discussion seems to go off-track.

Authors Response: The sentence was about our study. This was about the forage peanut selection by grazing animals. As we said in the last question, we were trying to say that although the legume proportion in the botanical composition was low, the animals selected the forage peanut to compose their diet. We could perform this affirmation by the isotopic 13C in the dung, which was numerically greater than in the botanical composition (but we did not perform the statistical analyses). Taking into account the digestibility of forage peanut, probably the legume in heifers’ diet was greater than in the botanical composition, which occurred by the animal selection. However, we removed the sentence to avoid confusion.

28. Since the proportions don’t change (except C5), you can say it in the text and no need for Fig. 3.

Authors Response: We are grateful for this comment. We removed the Fig. 3 and wrote the information in the results section: “The botanical composition of forage peanut and palisadegrass in the mixed treatment differed among GC (P < 0.001), however, the proportions did not change, except in GC5, which had the lowest forage peanut proportion (2% of the botanical composition). The proportion of forage peanut among GC1 to GC4, ranged between 7 to 9% of the botanical composition” (lines 343-347). 

29. This sentence does not sound correct. What do you mean?

Authors Response: Thank you for this comment. It is about the amount of herbage mass aboveground, which will affect the biological N2 fixation. We removed the sentence in the new manuscript version.

30. How many stages are there and what is the duration of these stages?

Authors Response: Thank you for this comment. In Alencar et al. (2018), the evaluations occurred only at the end of each experimental period, i.e., it was seven and eight months after seeding. However, the sentence was removed from the new manuscript version.

Alencar NM, Vendramini JMB, Santos AC, Silveira ML, Dubeux JCB, Sousa LF, et al. Herbage characteristics of pintoi peanut and palisadegrass established as monoculture or mixed swards. Crop Sci. 2018; 58:1–7. doi: http://dx.doi.org/10.2135/cropsci2017.09.0538

---

## [Decision Letter · Decision Letter 1]

13 Jan 2021

PONE-D-20-17268R1

Could forage peanut in low proportion replace N fertilizer in livestock systems?

PLOS ONE

Dear Dr. Longhini,

Thank you for submitting your manuscript to PLOS ONE. After careful consideration, we feel that it has merit but does not fully meet PLOS ONE’s publication criteria as it currently stands. Therefore, we invite you to submit a revised version of the manuscript that addresses the points raised during the review process.

The manuscript is much improved, and the revisions requested by the reviewers are relatively minor. I hope that both points made by reviewers are helpful, as indeed the study needs to only report mean separation where an ANOVA justify these being conducted. Also the abstract should only report on what the study evaluated, not the earlier over ambitous objectives before the study had to be revised based on modest growth observed. We look forward to your revised manuscript addressing these points.

We look forward to receiving your revised manuscript.

Kind regards,

Sieglinde S. Snapp

Academic Editor

PLOS ONE

Additional Editor Comments (if provided):

The manuscript is much improved and needs modest further inputs in order to be publishable, as indicated by both reviewers. The abstract and conclusion need to be revised to only report on the key findings and not beyond these, and similarly, mean separation should only be conducted if an anova is significant so this section needs to be updated to reflect this.

Reviewers' comments:

Reviewer's Responses to Questions

**Comments to the Author**

1. If the authors have adequately addressed your comments raised in a previous round of review and you feel that this manuscript is now acceptable for publication, you may indicate that here to bypass the “Comments to the Author” section, enter your conflict of interest statement in the “Confidential to Editor” section, and submit your "Accept" recommendation.

Reviewer #1: (No Response)

Reviewer #2: All comments have been addressed

2. Is the manuscript technically sound, and do the data support the conclusions?

Reviewer #1: Yes

Reviewer #2: Yes

3. Has the statistical analysis been performed appropriately and rigorously? 

Reviewer #1: Yes

Reviewer #2: No

4. Have the authors made all data underlying the findings in their manuscript fully available?

Reviewer #1: Yes

Reviewer #2: Yes

5. Is the manuscript presented in an intelligible fashion and written in standard English?

Reviewer #1: Yes

Reviewer #2: Yes

6. Review Comments to the Author

Reviewer #1: Suggestions for revision are made in the attached pdf file. Most are editorial in nature. Perhaps the one of greatest consequence relates to the last sentence of the Abstract and similar comments in the Conclusion section. These comments address treatments that were not tested in this study, so I don't think they really belong in these critical parts of your document.

Reviewer #2: The response to reviewer comments were thorough and the revised manuscript is a big improvement from the original submission. There remain multiple edits that need to be addressed by the authors prior to publication. Review the rules on when means separation tests can be used (i.e., only when the ANOVA test is significant).

7. PLOS authors have the option to publish the peer review history of their article (what does this mean?). If published, this will include your full peer review and any attached files.

Reviewer #1: No

Reviewer #2: No

---

## [Author Response · Author response to Decision Letter 1]

15 Feb 2021

Jaboticabal, Sao Paulo State, Brazil, February 15th, 2021.

Dear PLOS One Editor,

Subject: Submission of a revised manuscript PONE-D-20-17268R1 entitled “Could forage peanut in low proportion replace N fertilizer in livestock systems?”.

Thank you for considering our manuscript “Could forage peanut in low proportion replace N fertilizer in livestock systems?” for publication. We carefully addressed each of the reviewers’ comments, and we hope that the changes will meet your expectations. Our answers are listed below (in red) and a version of the revised manuscript with all the modifications (i.e., corrections and additional information inserted) highlighted in “track-change mode”. We would like to thank you and the reviewers for the excellent review, which certainly improved the manuscript quality. 

Thank you for the suggestions and possible appreciation of our work for publication in this renowned journal. The authors are available for other modifications that may improve the quality of the manuscript, as well as additional clarifications.

Sincerely,

Vanessa Zirondi Longhini, Ph.D.

Department of Animal Science

São Paulo State University (UNESP-FCAV)

Jaboticabal (SP), Zip 14884-900, Brazil

Comments to Author

Additional Editor Comments:

1. The manuscript is much improved and needs modest further inputs in order to be publishable, as indicated by both reviewers. The abstract and conclusion need to be revised to only report on the key findings and not beyond these, and similarly, mean separation should only be conducted if an anova is significant so this section needs to be updated to reflect this. 

Authors Response: Many thanks for considering a new version of the manuscript. We revised all your concerns and hope that all changes will meet your expectations. We modified the abstract, conclusion, and statistical analyzes to fit in the suggestions. 

Reviewers' comments to author

Reviewer 1

Suggestions for revision are made in the attached pdf file. Most are editorial in nature. Perhaps the one of greatest consequence relates to the last sentence of the Abstract and similar comments in the Conclusion section. These comments address treatments that were not tested in this study, so I don't think they really belong in these critical parts of your document.

Authors Response: Thank you very much for your effort in revising our manuscript. We agree with you on all points raised during the revision process. We removed the sentence from the abstract and conclusion section. All suggestion made in the pdf file was included in the new version.

1. Often, oftentimes is a colloquial expression.

Authors Response: Corrected. 

2. I don't believe you studied either of these treatments, so I don't think you want this to be the conclusion to your experiment. What did you find out relative to your objective?

Authors Response: The sentence was removed. We rephrase the sentence: “Overall, benefits from forage peanut were minimal because legume percentage was less than 10%, while N input in the system by N-fertilizer increased palisadegrass herbage biomass”. 

3. Where?? In this sentence you are talking about North America and one presumes a temperate environment, but the references are from warm North American climates. Please clarify where this further study is needed.

Authors Response: The studies occurred in Florida. We rephrased the sentence: “In the Southeastern United States, mixing grasses and legumes is a more common practice [11] that requires further study [12, 13]”.

4. This is not Mehlich exponent -1, it is Mehlich-1. Mehlich-1 is a type of extractant used in soil testing.

Authors Response: Corrected. 

5. The first sentence is methods. In the second sentence, do we really want to be talking about a "difference" with a P value of 0.14. I really don't think so.

Authors Response: Thank you for catching it. We removed both sentences. 

6. when you say cell production, do you mean cell division or elongation, or both?

Authors Response: I was referring to cell division. According to Volenec and Nelson (1983) leaf elongation rate was affected by N fertilizer, which may have been linked to cell division, while cell length remains constant.

Volenec JJ, Nelson CJ. Responses of Tall Fescue Leaf Meristems to N Fertilization and Harvest Frequency 1. Crop Sci. 1983; 23:720–724. doi: 10.2135/cropsci1983.0011183X002300040028x

7. clarify here that you are talking about secondary cell walls in stems, not leaves.

Authors Response: Thank you for this comment. We rephrased the sentence to clarify: “Stem acts as a leaf support structure with lignified secondary cell and other less digestible components [40, 41], providing stiffness to the stem. Combined, these traits may reduce animal performance [42]”.

8. Upon what do you base this conclusion since you did not evaluate different levels of LI?

Authors Response: We agree with your comment. We did not evaluate the light interception level. We rephrase the sentence to clarify: “However, in our study with a minimal proportion of forage peanut, it may be necessary to keep the LI less than 90% in order to reduce grass competition with the forage peanut [47]”. The information about 90% LI was removed from the abstract and conclusion section. 

9. or perhaps reduce legume proportion due to greater competition from the grass??

Authors Response: The sentence was rephrased to clarify: “Further studies are required to evaluate whether N fertilization applied at key periods (when conditions are more favorable to forage peanut growth than grass growth) could hasten the legume proportion, at least during the initial years. These factors require further research in pursuit of developing sustainable mixed systems that provide quality feed to grazing animals, provide cost reduction to the farmer and lessen the environmental impacts of synthetic N fertilizers”. 

Reviewer 2

The response to reviewer comments were thorough and the revised manuscript is a big improvement from the original submission. There remain multiple edits that need to be addressed by the authors prior to publication. Review the rules on when means separation tests can be used (i.e., only when the ANOVA test is significant). 

Authors Response: Thank you very much for all the valuable advice and points raised during the revision process. We hope that all changes will meet your expectations. We included the suggestions made in both files.

1. In all cases, it should be Mehlich-1 (no superscripts).

Authors Response: Corrected.

2. To verify that it was P and not P2O5. To verify it was K and not K2O.

Authors Response: Thank you for catching it. We used the amount of phosphate (40 kg P ha-1) and chloride (50 kg K ha-1) for fertilizer calculation. Then we used 83.3 kg K2O ha-1 and 222.2 kg P2O5 ha-1. We rephrased the sentence to clarify: At the beginning of the grazing trial in both years, 40 kg P ha-1 (equivalent to 222.2 kg P2O5 ha-1) and 50 kg K ha-1 (equivalent to 83.3 kg K2O ha-1) were applied.

3. include a more complete address.

Authors Response: We included the complete address: “Embrapa Agrobiologia, Seropedica, RJ, Brazil”.

4. You cannot use means separation if the ANOVA is not significant. 0.104 and 0.143 are not significant, so do not run means separation (no letters for CH or FP).

Authors Response: Thank you for catching it. We removed this information from the revised manuscript.

5. Not sure what the journal rules are, but typically one will list the definitions in the same order as they are presented in the table.

Authors Response: We revised according to journal rules.

6. I do not understand this sentence. I think of botanical composition as proportions, but you follow that proportions only changed at GC 5.

Authors Response: The botanical composition included the proportion of the palisadegrass and forage peanut in the mixed treatment. The figure about botanical composition was removed as suggested in the last review process, including only the text in the results section. We rephrased the sentence to clarify: “The botanical composition of forage peanut and palisadegrass in the mixed treatment differed among GCs (P < 0.001). However, these proportions did not change among GC1 to GC4 (ranging between 7 to 9% of the legume in the botanical composition), except in GC5, which had the least forage peanut proportion (2% of the botanical composition)”.

7. This reasoning is difficult to follow and you do not explain why the reader should even be interested. So what? Don't fixate on explaining each response but rather what is its contribution to the overall performance of the system? Is taller canopy a benefit or simply a reflection of greater biomass, in general? If you have greater tiller mass, then you can assume the plant is increasing its ground cover, which is beneficial. Also, maybe there is a relationship between greater tiller mass to L:S ratios. Again, it needs to mean something. Just because you measured a difference does not necessarily mean it is a meaningful part of the story.

Authors Response: We agree with your comment. The sentence was removed from the new version.

8. You have more dead biomass with the N treatment because you have more living biomass with the N treatment. You need to compare the ratio of senescent to living. If you do that, you will find that actually, the N fertilizer treatment had a much lower ratio of dead to living. That makes total sense, as often, low N fertilization actually hastens plant life cycles and there is greater senescence.

Authors Response: We agree with your comment. The senescent proportion represented 24% of total biomass in the N fertilizer treatment, while mixed and control treatment represented 34 and 31%, respectively. We included the sentence: “There was more senescent biomass in the N fertilized treatment because there was more living biomass (leaf + stem). However, when senescent biomass was compared to the living biomass as a ratio, the N fertilizer treatment had a much lower proportion of senescent to living biomass, representing only 24% of the total herbage biomass. In contrast, in the unfertilized and mixed treatments, the senescent biomass represented 31 and 34% of the total herbage mass, respectively. The low N fertilization may have hastened the plant growth cycle which promoted greater senescence”.

9. This has little to do with your results. Your results are that the forage peanut did poorly, but it sounds as though you are daydreaming about what it would be like if the forage peanut performed well. Stick with your results!

Authors Response: We agree with your comment. We removed the sentence from the revised manuscript.

10. It is well known that if you apply N fertilizer, you will make the grass competition that much worse. Perhaps it would help only if applied at the beginning of the rainy season, when the forage peanut is already outgrowing (based on what you reported earlier) the grass. Applying N mid-season will likely only favor the grass (many reports on grass/legume competition).

Authors Response: We agree with your comment. We removed the sentence from the revised manuscript.

11. Requires a higher resolution image. This one is blurry. This figure is real helpful for understanding the grazing management.

Authors Response: We appreciate this comment. We made the changes as suggested. Please, download the image from the PDF document to see the improved resolution in the image.

12. Would be better if fertilized or mixed legend used a pattern instead of a shade, as it is difficult to distinguish between the 2 treatments. Fig. 3: In the figure caption, you need to explain what the data bars represent (mean), along with the error bars and letters. Fig3 b you need to explain the asterisk.

Authors Response: We appreciate this comment. We changed the fertilized legend to black and mixed legend to white. The complete figure caption was described in the manuscript. Please, see lines 129-136, 249-257, and 289-294 in the “Revised Manuscript with Track Changes file”.

13. I do not see any comparisons among GCs even though it was highly significant.

Authors Response: Thank you for your comment. We included the GC means in Figure 4.

---

## [Editor Report · Decision Letter 2]

17 Feb 2021

Could forage peanut in low proportion replace N fertilizer in livestock systems?

PONE-D-20-17268R2

Dear Dr. Longhini,

We’re pleased to inform you that your manuscript has been judged scientifically suitable for publication and will be formally accepted for publication once it meets all outstanding technical requirements.

Kind regards,

Sieglinde S. Snapp

Academic Editor

PLOS ONE

Additional Editor Comments (optional):

The authors have done an excellent job of addressing all remaining issues.

---

## [Editor Report · Acceptance letter]

22 Feb 2021

PONE-D-20-17268R2 

Could forage peanut in low proportion replace N fertilizer in livestock systems? 

Dear Dr. Longhini:

I'm pleased to inform you that your manuscript has been deemed suitable for publication in PLOS ONE. Congratulations! Your manuscript is now with our production department. 

Kind regards, 

on behalf of

Dr. Sieglinde S. Snapp 

Academic Editor

PLOS ONE